# Convergent Policy Optimization for Safe Reinforcement Learning

**Ming Yu** [*]     **Zhuoran Yang** [†]     **Mladen Kolar** [‡]     **Zhaoran Wang** [§]

## Abstract

We study the safe reinforcement learning problem with nonlinear function approx-
imation, where policy optimization is formulated as a constrained optimization
problem with both the objective and the constraint being nonconvex functions.
For such a problem, we construct a sequence of surrogate convex constrained
optimization problems by replacing the nonconvex functions locally with convex
quadratic functions obtained from policy gradient estimators. We prove that the
solutions to these surrogate problems converge to a stationary point of the original
nonconvex problem. Furthermore, to extend our theoretical results, we apply our
algorithm to examples of optimal control and multi-agent reinforcement learning
with safety constraints.

## 1   Introduction

Reinforcement learning [58] has achieved tremendous success in video games [41, 44, 56, 36, 66]
and board games, such as chess and Go [53, 55, 54], in part due to powerful simulators [8, 62]. In
contrast, due to physical limitations, real-world applications of reinforcement learning methods often
need to take into consideration the safety of the agent [5, 26]. For instance, in expensive robotic
and autonomous driving platforms, it is pivotal to avoid damages and collisions [25, 9]. In medical
applications, we need to consider the switching cost [7].

A popular model of safe reinforcement learning is the constrained Markov decision process (CMDP),
which generalizes the Markov decision process by allowing for inclusion of constraints that model
the concept of safety [3]. In a CMDP, the cost is associated with each state and action experienced
by the agent, and safety is ensured only if the expected cumulative cost is below a certain threshold.
Intuitively, if the agent takes an unsafe action at some state, it will receive a huge cost that punishes
risky attempts. Moreover, by considering the cumulative cost, the notion of safety is defined for the
whole trajectory enabling us to examine the long-term safety of the agent, instead of focusing on
individual state-action pairs. For a CMDP, the goal is to take sequential decisions to achieve the
expected cumulative reward under the safety constraint.

Solving a CMDP can be written as a linear program [3], with the number of variables being the
same as the size of the state and action spaces. Therefore, such an approach is only feasible for the
tabular setting, where we can enumerate all the state-action pairs. For large-scale reinforcement
learning problems, where function approximation is applied, both the objective and constraint of the
CMDP are nonconvex functions of the policy parameter. One common method for solving CMDP
is to formulate an unconstrained saddle-point optimization problem via Lagrangian multipliers and
solve it using policy optimization algorithms [18, 60]. Such an approach suffers the following two
drawbacks:

---

[*]The University of Chicago Booth School of Business, Chicago, IL. Email: `ming93@uchicago.edu`.

[†]Department of Operations Research and Financial Engineering, Princeton University, Princeton, NJ.

[‡]The University of Chicago Booth School of Business, Chicago, IL.

[§]Department of Industrial Engineering and Management Sciences, Northwestern University, Evanston, IL.

First, for each fixed Lagrangian multiplier, the inner minimization problem itself can be viewed as solving a new reinforcement learning problem. From the computational point of view, solving the saddle-point optimization problem requires solving a sequence of MDPs with different reward functions. For a large scale problem, even solving a single MDP requires huge computational resources, making such an approach computationally infeasible.

Second, from a theoretical perspective, the performance of the saddle-point approach hinges on solving the inner problem optimally. Existing theory only provides convergence to a stationary point where the gradient with respect to the policy parameter is zero [28, 37]. Moreover, the objective, as a bivariate function of the Lagrangian multiplier and the policy parameter, is not convex-concave and, therefore, first-order iterative algorithms can be unstable [27].

In contrast, we tackle the nonconvex constrained optimization problem of the CMDP directly. We propose a novel policy optimization algorithm, inspired by [38]. Specifically, in each iteration, we replace both the objective and constraint by quadratic surrogate functions and update the policy parameter by solving the new constrained optimization problem. The two surrogate functions can be viewed as first-order Taylor-expansions of the expected reward and cost functions where the gradients are estimated using policy gradient methods [59]. Additionally, they can be viewed as convex relaxations of the original nonconvex reward and cost functions. In §4 we show that, as the algorithm proceeds, we obtain a sequence of convex relaxations that gradually converge to a smooth function. More importantly, the sequence of policy parameters converges almost surely to a stationary point of the nonconvex constrained optimization problem.

**Related work.** Our work is pertinent to the line of research on CMDP [3]. For CMDPs with large state and action spaces, [19] proposed an iterative algorithm based on a novel construction of Lyapunov functions. However, their theory only holds for the tabular setting. Using Lagrangian multipliers, [46, 18, 1, 60] proposed policy gradient [59], actor-critic [33], or trust region policy optimization [51] methods for CMDP or constrained risk-sensitive reinforcement learning [26]. These algorithms either do not have convergence guarantees or are shown to converge to saddle-points of the Lagrangian using two-time-scale stochastic approximations [10]. However, due to the projection on the Lagrangian multiplier, the saddle-point achieved by these approaches might not be the stationary point of the original CMDP problem. In addition, [65] proposed a cross-entropy-based stochastic optimization algorithm, and proved the asymptotic behavior using ordinary differential equations. In contrast, our algorithm and the theoretical analysis focus on the discrete time CMDP. Outside of the CMDP setting, [31, 35] studied safe reinforcement learning with demonstration data, [61] studied the safe exploration problem with different safety constraints, and [4] studied multi-task safe reinforcement learning.

**Our contribution.** Our contribution is three-fold. First, for the CMDP policy optimization problem where both the objective and constraint function are nonconvex, we propose to optimize a sequence of convex relaxation problems using convex quadratic functions. Solving these surrogate problems yields a sequence of policy parameters that converge almost surely to a stationary point of the original policy optimization problem. Second, to reduce the variance in the gradient estimator that is used to construct the surrogate functions, we propose an online actor-critic algorithm. Finally, as concrete applications, our algorithms are also applied to optimal control (in §5) and parallel and multi-agent reinforcement learning problems with safety constraints (in supplementary material).

## 2 Background

A Markov decision process is denoted by $(\mathcal{S}, \mathcal{A}, P, \gamma, r, \mu)$, where $\mathcal{S}$ is the state space, $\mathcal{A}$ is the action space, $P$ is the transition probability distribution, $\gamma \in (0, 1)$ is the discount factor, $r \colon \mathcal{S} \times \mathcal{A} \to \mathbb{R}$ is the reward function, and $\mu \in \mathcal{P}(\mathcal{S})$ is the distribution of the initial state $s_0 \in \mathcal{S}$, where we denote $\mathcal{P}(\mathcal{X})$ as the set of probability distributions over $\mathcal{X}$ for any $\mathcal{X}$. A policy is a mapping $\pi : \mathcal{S} \to \mathcal{P}(\mathcal{A})$ that specifies the action that an agent will take when it is at state $s$.

**Policy gradient method.** Let $\{\pi_\theta \colon \mathcal{S} \to \mathcal{P}(\mathcal{A})\}$ be a parametrized policy class, where $\theta \in \Theta$ is the parameter defined on a compact set $\Theta$. This parameterization transfers the original infinite dimensional policy class to a finite dimensional vector space and enables gradient based methods to be used to maximize (1). For example, the most popular Gaussian policy can be written as $\pi(\cdot|s, \theta) = \mathcal{N}\big(\mu(s, \theta), \sigma(s, \theta)\big)$, where the state dependent mean $\mu(s, \theta)$ and standard deviation

$\sigma(s, \theta)$ can be further parameterized as $\mu(s, \theta) = \theta_\mu^\top \cdot x(s)$ and $\sigma(s, \theta) = \exp\left(\theta_\sigma^\top \cdot x(s)\right)$ with $x(s)$ being a state feature vector. The goal of an agent is to maximize the expected cumulative reward

$$R(\theta) = \mathbb{E}_\pi \left[ \sum_{t \geq 0} \gamma^t \cdot r(s_t, a_t) \right], \tag{1}$$

where $s_0 \sim \mu$, and for all $t \geq 0$, we have $s_{t+1} \sim P(\cdot \,|\, s_t, a_t)$ and $a_t \sim \pi(\cdot \,|\, s_t)$. Given a policy $\pi(\theta)$, we define the state- and action-value functions of $\pi_\theta$, respectively, as

$$V^\theta(s) = \mathbb{E}_{\pi_\theta} \left[ \sum_{t \geq 0} \gamma^t r(s_t, a_t) \,\middle|\, s_0 = s \right], \; Q^\theta(s, a) = \mathbb{E}_{\pi_\theta} \left[ \sum_{t \geq 0} \gamma^t r(s_t, a_t) \,\middle|\, s_0 = s, a_0 = a \right]. \tag{2}$$

The policy gradient method updates the parameter $\theta$ through gradient ascent

$$\theta_{k+1} = \theta_k + \eta \cdot \widehat{\nabla}_\theta R(\theta_k),$$

where $\widehat{\nabla}_\theta R(\theta_k)$ is a stochastic estimate of the gradient $\nabla_\theta R(\theta_k)$ at $k$-th iteration. Policy gradient method, as well as its variants (e.g. policy gradient with baseline [58], neural policy gradient [64, 39, 16]) is widely used in reinforcement learning. The gradient $\nabla_\theta R(\theta)$ can be estimated according to the policy gradient theorem [59],

$$\nabla_\theta R(\theta) = \mathbb{E}\left[ \nabla_\theta \log \pi_\theta(s, a) \cdot Q^\theta(s, a) \right]. \tag{3}$$

**Actor-critic method.** To further reduce the variance of the policy gradient method, we could estimate both the policy parameter and value function simultaneously. This kind of method is called actor-critic algorithm [33], which is widely used in reinforcement learning. Specifically, in the value function evaluation (*critic*) step we estimate the action-value function $Q^\theta(s, a)$ using, for example, the temporal difference method TD(0) [20]. The policy parameter update (*actor*) step is implemented as before by the Monte-Carlo method according to the policy gradient theorem (3) with the action-value $Q^\theta(s, a)$ replaced by the estimated value in the policy evaluation step.

**Constrained MDP.** In this work, we consider an MDP problem with an additional constraint on the model parameter $\theta$. Specifically, when taking action at some state we incur some cost value. The constraint is such that the expected cumulative cost cannot exceed some pre-defined constant. A constrained Markov decision process (CMDP) is denoted by $(\mathcal{S}, \mathcal{A}, P, \gamma, r, d, \mu)$, where $d\colon \mathcal{S} \times \mathcal{A} \to \mathbb{R}$ is the cost function and the other parameters are as before. The goal of an the agent in CMDP is to solve the following constrained problem

$$\begin{aligned} \underset{\theta \in \Theta}{\text{minimize}} \quad & J(\theta) = \mathbb{E}_{\pi_\theta}\left[ -\sum_{t \geq 0} \gamma^t \cdot r(s_t, a_t) \right], \\ \text{subject to} \quad & D(\theta) = \mathbb{E}_{\pi_\theta}\left[ \sum_{t \geq 0} \gamma^t \cdot d(s_t, a_t) \right] \leq D_0, \end{aligned} \tag{4}$$

where $D_0$ is a fixed constant. We consider only one constraint $D(\theta) \leq D_0$, noting that it is straightforward to generalize to multiple constraints. Throughout this paper, we assume that both the reward and cost value functions are bounded: $\left| r(s_t, a_t) \right| \leq r_{\max}$ and $\left| d(s_t, a_t) \right| \leq d_{\max}$. Also, the parameter space $\Theta$ is assumed to be compact.

## 3 Algorithm

In this section, we develop an algorithm to solve the optimization problem (4). Note that both the objective function and the constraint in (4) are nonconvex and involve expectation without closed-form expression. As a constrained problem, a straightforward approach to solve (4) is to define the following Lagrangian function

$$L(\theta, \lambda) = J(\theta) + \lambda \cdot \left[ D(\theta) - D_0 \right],$$

and solve the dual problem

$$\inf_{\lambda \geq 0} \sup_\theta L(\theta, \lambda).$$

However, this problem is a nonconvex minimax problem and, therefore, is hard to solve and establish theoretical guarantees for solutions [2]. Another approach to solve (4) is to replace $J(\theta)$ and $D(\theta)$ by surrogate functions with nice properties. For example, one can iteratively construct local quadratic approximations that are strongly convex [52], or are an upper bound for the original function [57]. However, an immediate problem of this naive approach is that, even if the original problem (4) is feasible, the convex relaxation problem need not be. Also, these methods only deal with deterministic and/or convex constraints.

In this work, we propose an iterative algorithm that approximately solves (4) by constructing a sequence of convex relaxations, inspired by [38]. Our method is able to handle the possible infeasible situation due to the convex relaxation as mentioned above, and handle stochastic and nonconvex constraint. Since we do not have access to $J(\theta)$ or $D(\theta)$, we first define the sample negative cumulative reward and cost functions as

$$J^*(\theta) = -\sum_{t \geq 0} \gamma^t \cdot r(s_t, a_t) \qquad \text{and} \qquad D^*(\theta) = \sum_{t \geq 0} \gamma^t \cdot d(s_t, a_t).$$

Given $\theta$, $J^*(\theta)$ and $D^*(\theta)$ are the sample negative cumulative reward and cost value of a realization (i.e., a trajectory) following policy $\pi_\theta$. Note that both $J^*(\theta)$ and $D^*(\theta)$ are stochastic due to the randomness in the policy, state transition distribution, etc. With some abuse of notation, we use $J^*(\theta)$ and $D^*(\theta)$ to denote both a function of $\theta$ and a value obtained by the realization of a trajectory. Clearly we have $J(\theta) = \mathbb{E}\big[J^*(\theta)\big]$ and $D(\theta) = \mathbb{E}\big[D^*(\theta)\big]$.

We start from some (possibly infeasible) $\theta_0$. Let $\theta_k$ denote the estimate of the policy parameter in the $k$-th iteration. As mentioned above, we do not have access to the expected cumulative reward $J(\theta)$. Instead we sample a trajectory following the current policy $\pi_{\theta_k}$ and obtain a realization of the negative cumulative reward value and the gradient of it as $J^*(\theta_k)$ and $\nabla_\theta J^*(\theta_k)$, respectively. The cumulative reward value is obtained by Monte-Carlo estimation, and the gradient is also obtained by Monte-Carlo estimation according to the policy gradient theorem in (3). We provide more details on the realization step later in this section. Similarly, we use the same procedure for the cost function and obtain realizations $D^*(\theta_k)$ and $\nabla_\theta D^*(\theta_k)$.

We approximate $J(\theta)$ and $D(\theta)$ at $\theta_k$ by the quadratic surrogate functions

$$\widetilde{J}(\theta, \theta_k, \tau) = J^*(\theta_k) + \langle \nabla_\theta J^*(\theta_k), \theta - \theta_k \rangle + \tau \|\theta - \theta_k\|_2^2, \tag{5}$$

$$\widetilde{D}(\theta, \theta_k, \tau) = D^*(\theta_k) + \langle \nabla_\theta D^*(\theta_k), \theta - \theta_k \rangle + \tau \|\theta - \theta_k\|_2^2, \tag{6}$$

where $\tau > 0$ is any fixed constant. In each iteration, we solve the optimization problem

$$\overline{\theta}_k = \underset{\theta}{\operatorname{argmin}} \, \overline{J}^{(k)}(\theta) \qquad \text{subject to} \qquad \overline{D}^{(k)}(\theta) \leq D_0, \tag{7}$$

where we define

$$\overline{J}^{(k)}(\theta) = (1 - \rho_k) \cdot \overline{J}^{(k-1)}(\theta) + \rho_k \cdot \widetilde{J}(\theta, \theta_k, \tau), \tag{8}$$

$$\overline{D}^{(k)}(\theta) = (1 - \rho_k) \cdot \overline{D}^{(k-1)}(\theta) + \rho_k \cdot \widetilde{D}(\theta, \theta_k, \tau),$$

with the initial value $\overline{J}^{(0)}(\theta) = \overline{D}^{(0)}(\theta) = 0$. Here $\rho_k$ is the weight parameter to be specified later. According to the definition (5) and (6), problem (7) is a convex quadratically constrained quadratic program (QCQP). Therefore, it can be efficiently solved by, for example, the interior point method. However, as mentioned before, even if the original problem (4) is feasible, the convex relaxation problem (7) could be infeasible. In this case, we instead solve the following feasibility problem

$$\overline{\theta}_k = \underset{\theta, \alpha}{\operatorname{argmin}} \, \alpha \qquad \text{subject to} \qquad \overline{D}^{(k)}(\theta) \leq D_0 + \alpha. \tag{9}$$

In particular, we relax the infeasible constraint and find $\overline{\theta}_k$ as the solution that gives the minimum relaxation. Due to the specific form in (6), $\overline{D}^{(k)}(\theta)$ is decomposable into quadratic forms of each component of $\theta$, with no terms involving $\theta_i \cdot \theta_j$. Therefore, the solution to problem (9) can be written in a closed form. Given $\overline{\theta}_k$ from either (7) or (9), we update $\theta_k$ by

$$\theta_{k+1} = (1 - \eta_k) \cdot \theta_k + \eta_k \cdot \overline{\theta}_k, \tag{10}$$

where $\eta_k$ is the learning rate to be specified later. Note that although we consider only one constraint in the algorithm, both the algorithm and the theoretical result in Section 4 can be directly generalized to multiple constraints setting. The whole procedure is summarized in Algorithm 1.

---
**Algorithm 1** Successive convex relaxation algorithm for constrained MDP
---
1: **Input:** Initial value $\theta_0$, $\tau$, $\{\rho_k\}$, $\{\eta_k\}$.
2: **for** $k = 1, 2, 3, \ldots$ **do**
3:     Obtain a sample $J^*(\theta_k)$ and $D^*(\theta_k)$ by Monte-Carlo sampling.
4:     Obtain a sample $\nabla_\theta J^*(\theta_k)$ and $\nabla_\theta D^*(\theta_k)$ by policy gradient theorem.
5:     **if** problem (7) is feasible **then**
6:         Obtain $\overline{\theta}_k$ by solving (7).
7:     **else**
8:         Obtain $\overline{\theta}_k$ by solving (9).
9:     **end if**
10:    Update $\theta_{k+1}$ by (10).
11: **end for**
---

**Obtaining realizations $J^*(\theta_k)$ and $\nabla_\theta J^*(\theta_k)$.** We detail how to obtain realizations $J^*(\theta_k)$ and $\nabla_\theta J^*(\theta_k)$ corresponding to the lines 3 and 4 in Algorithm 1. The realizations of $D^*(\theta_k)$ and $\nabla_\theta D^*(\theta_k)$ can be obtained similarly.

First, we discuss finite horizon setting, where we can sample the full trajectory according to the policy $\pi_\theta$. In particular, for any $\theta_k$, we use the policy $\pi_{\theta_k}$ to sample a trajectory and obtain $J^*(\theta_k)$ by Monte-Carlo method. The gradient $\nabla_\theta J(\theta)$ can be estimated by the policy gradient theorem [59],

$$\nabla_\theta J(\theta) = -\mathbb{E}_{\pi_\theta}\Big[\nabla_\theta \log \pi_\theta(s, a) \cdot Q^\theta(s, a)\Big]. \tag{11}$$

Again we can sample a trajectory and obtain the policy gradient realization $\nabla_\theta J^*(\theta_k)$ by Monte-Carlo method.

In infinite horizon setting, we cannot sample the infinite length trajectory. In this case, we utilize the truncation method introduced in [48], which truncates the trajectory at some stage $T$ and scales the undiscounted cumulative reward to obtain an unbiased estimation. Intuitively, if the discount factor $\gamma$ is close to 0, then the future reward would be discounted heavily and, therefore, we can obtain an accurate estimate with a relatively small number of stages. On the other hand, if $\gamma$ is close to 1, then the future reward is more important compared to the small $\gamma$ case and we have to sample a long trajectory. Taking this intuition into consideration, we define $T$ to be a geometric random variable with parameter $1 - \gamma$: $\Pr(T = t) = (1 - \gamma)\gamma^t$. Then, we simulate the trajectory until stage $T$ and use the estimator $J_{\text{truncate}}(\theta) = -(1 - \gamma) \cdot \sum_{t=0}^{T} r(s_t, a_t)$, which is an unbiased estimator of the expected negative cumulative reward $J(\theta)$, as proved in proposition 5 in [43]. We can apply the same truncation procedure to estimate the policy gradient $\nabla_\theta J(\theta)$.

**Variance reduction.** Using the naive sampling method described above, we may suffer from high variance problem. To reduce the variance, we can modify the above procedure in the following ways. First, instead of sampling only one trajectory in each iteration, a more practical and stable way is to sample several trajectories and take average to obtain the realizations. As another approach, we can subtract a baseline function from the action-value function $Q^\theta(s, a)$ in the policy gradient estimation step (11) to reduce the variance without changing the expectation. A popular choice of the baseline function is the state-value function $V^\theta(s)$ as defined in (2). In this way, we can replace $Q^\theta(s, a)$ in (11) by the advantage function $A^\theta(s, a)$ defined as

$$A^\theta(s, a) = Q^\theta(s, a) - V^\theta(s).$$

This modification corresponds to the standard REINFORCE with Baseline algorithm [58] and can significantly reduce the variance of policy gradient.

**Actor-critic method.** Finally, we can use an actor-critic update to improve the performance further. In this case, since we need unbiased estimators for both the gradient and the reward value in (5) and (6) in online fashion, we modify our original problem (4) to average reward setting as

$$\underset{\theta \in \Theta}{\text{minimize}} \quad J(\theta) = \lim_{T \to \infty} \mathbb{E}_{\pi_\theta}\left[-\frac{1}{T} \sum_{t=0}^{T} r(s_t, a_t)\right],$$

$$\text{subject to} \quad D(\theta) = \lim_{T \to \infty} \mathbb{E}_{\pi_\theta}\left[\frac{1}{T} \sum_{t=0}^{T} d(s_t, a_t)\right] \leq D_0.$$

---

**Algorithm 2** Actor-Critic update for constrained MDP

---

1: **for** $k = 1, 2, 3, \ldots$ **do**
2:     Take action $a$, observe reward $r$, cost $d$, and new state $s'$.
3:     **Critic step:**
4:         $w \leftarrow w + \beta_w \cdot \delta^J \nabla_w V_w^J(s), \;\; J \leftarrow J + \beta_w \cdot (r - J)$.
5:         $v \leftarrow v + \beta_v \cdot \delta^D \nabla_v V_v^J(s), \;\; D \leftarrow D + \beta_v \cdot (d - D)$.
6:     **Calculate TD error:**
7:         $\delta^J = r - J + V_w^J(s') - V_w^J(s)$.
8:         $\delta^D = d - D + V_v^D(s') - V_v^D(s)$.
9:     **Actor step:**
10:       Solve $\bar{\theta}_k$ by (7) or (9) with
           $J^*(\theta_k)$, $\nabla_\theta J^*(\theta_k)$ in (5) replaced by $J$ and $\delta^J \cdot \nabla_\theta \log \pi_\theta(s, a)$;
           $D^*(\theta_k)$, $\nabla_\theta D^*(\theta_k)$ in (6) replaced by $D$ and $\delta^D \cdot \nabla_\theta \log \pi_\theta(s, a)$.
11:     $s \leftarrow s'$.
12: **end for**

---

Let $V_\theta^J(s)$ and $V_\theta^D(s)$ denote the value and cost functions corresponding to (2). We use possibly nonlinear approximation with parameter $w$ for the value function: $V_w^J(s)$ and $v$ for the cost function: $V_v^D(s)$. In the critic step, we update $w$ and $v$ by TD(0) with step size $\beta_w$ and $\beta_v$; in the actor step, we solve our proposed convex relaxation problem to update $\theta$. The actor-critic procedure is summarized in Algorithm 2. Here $J$ and $D$ are estimators of $J(\theta_k)$ and $D(\theta_k)$. Both of $J$ and $D$, and the TD error $\delta^J$, $\delta^D$ can be initialized as 0.

The usage of the actor-critic method helps reduce variance by using a value function instead of Monte-Carlo sampling. Specifically, in Algorithm 1 we need to obtain a sample trajectory and calculate $J^*(\theta)$ and $\nabla_\theta J^*(\theta)$ by Monte-Carlo sampling. This step has a high variance since we need to sample a potentially long trajectory and sum up a lot of random rewards. In contrast, in Algorithm 2, this step is replaced by a value function $V_w^J(s)$, which reduces the variance.

## 4 Theoretical Result

In this section, we show almost sure convergence of the iterates obtained by our algorithm to a stationary point. We start by stating some mild assumptions on the original problem (4) and the choice of some parameters in Algorithm 1.

**Assumption 1** *The choice of $\{\eta_k\}$ and $\{\rho_k\}$ satisfy $\lim_{k \to \infty} \sum_k \eta_k = \infty$, $\lim_{k \to \infty} \sum_k \rho_k = \infty$ and $\lim_{k \to \infty} \sum_k \eta_k^2 + \rho_k^2 < \infty$. Furthermore, we have $\lim_{k \to \infty} \eta_k / \rho_k = 0$ and $\eta_k$ is decreasing.*

**Assumption 2** *For any realization, $J^*(\theta)$ and $D^*(\theta)$ are continuously differentiable as functions of $\theta$. Moreover, $J^*(\theta)$, $D^*(\theta)$, and their derivatives are uniformly Lipschitz continuous.*

Assumption 1 allows us to specify the learning rates. A practical choice would be $\eta_k = k^{-c_1}$ and $\rho_k = k^{-c_2}$ with $0.5 < c_2 < c_1 < 1$. This assumption is standard for gradient-based algorithms. Assumption 2 is also standard and is known to hold for a number of models. It ensures that the reward and cost functions are sufficiently regular. In fact, it can be relaxed such that each realization is Lipschitz (not uniformly), and the event that we keep generating realizations with monotonically increasing Lipschitz constant is an event with probability 0. See condition iv) in [67] and the discussion thereafter. Also, see [45] for sufficient conditions such that both the expected cumulative reward function and the gradient of it are Lipschitz.

The following Assumption 3 is useful only when we initialize with an infeasible point. We first state it here and we will discuss this assumption after the statement of the main theorem.

**Assumption 3** *Suppose $(\theta_S, \alpha_S)$ is a stationary point of the optimization problem*

$$\underset{\theta, \alpha}{minimize} \;\; \alpha \qquad subject\ to \qquad D(\theta) \leq D_0 + \alpha. \tag{12}$$

*We have that $\theta_S$ is a feasible point of the original problem (4), i.e. $D(\theta_S) \leq D_0$.*

We are now ready to state the main theorem.

**Theorem 4** *Suppose the Assumptions 1 and 2 are satisfied with small enough initial step size $\eta_0$. Suppose also that, either $\theta_0$ is a feasible point, or Assumption 3 is satisfied. If there is a subsequence $\{\theta_{k_j}\}$ of $\{\theta_k\}$ that converges to some $\widetilde{\theta}$, then there exist uniformly continuous functions $\widehat{J}(\theta)$ and $\widehat{D}(\theta)$ satisfying*

$$\lim_{j \to \infty} \overline{J}^{(k_j)}(\theta) = \widehat{J}(\theta) \qquad and \qquad \lim_{j \to \infty} \overline{D}^{(k_j)}(\theta) = \widehat{D}(\theta).$$

*Furthermore, suppose there exists $\theta$ such that $\widehat{D}(\theta) < D_0$ (i.e. the Slater's condition holds), then $\widetilde{\theta}$ is a stationary point of the original problem* (4) *almost surely.*

The proof of Theorem 4 is provided in the supplementary material.

Note that Assumption 3 is not necessary if we start from a feasible point, or we reach a feasible point in the iterates, which could be viewed as an initializer. Assumption 3 makes sure that the iterates in Algorithm 1 keep making progress without getting stuck at any infeasible stationary point. A similar condition is assumed in [38] for an infeasible initializer. If it turns out that $\theta_0$ is infeasible and Assumption 3 is violated, then the convergent point may be an infeasible stationary point of (12). In practice, if we can find a feasible point of the original problem, then we proceed with that point. Alternatively, we could generate multiple initializers and obtain iterates for all of them. As long as there is a feasible point in one of the iterates, we can view this feasible point as the initializer and Theorem 4 follows without Assumption 3. In our later experiments, for every single replicate, we could reach a feasible point, and therefore Assumption 3 is not necessary.

Our algorithm does not guarantee safe exploration during the training phase. Ensuring safety during learning is a more challenging problem. Sometimes even finding a feasible point is not straightforward, otherwise Assumption 3 is not necessary.

Our proposed algorithm is inspired by [38]. Compared to [38] which deals with an optimization problem, solving the safe reinforcement learning problem is more challenging. We need to verify that the Lipschitz condition is satisfied, and also the policy gradient has to be estimated (instead of directly evaluated as in a standard optimization problem). The usage of the Actor-Critic algorithm reduces the variance of the sampling, which is unique to Reinforcement learning.

## 5    Application to Constrained Linear-Quadratic Regulator

We apply our algorithm to the linear-quadratic regulator (LQR), which is one of the most fundamental problems in control theory. In the LQR setting, the state dynamic equation is linear, the cost function is quadratic, and the optimal control theory tells us that the optimal control for LQR is a linear function of the state [23, 6]. LQR can be viewed as an MDP problem and it has attracted a lot of attention in the reinforcement learning literature [12, 13, 21, 47].

We consider the infinite-horizon, discrete-time LQR problem. Denote $x_t$ as the state variable and $u_t$ as the control variable. The state transition and the control sequence are given by

$$\begin{aligned} x_{t+1} &= Ax_t + Bu_t + v_t, \\ u_t &= -Fx_t + w_t, \end{aligned} \tag{13}$$

where $v_t$ and $w_t$ represent possible Gaussian white noise, and the initial state is given by $x_0$. The goal is to find the control parameter matrix $F$ such that the expected total cost is minimized. The usual cost function of LQR corresponds to the negative reward in our setting and we impose an additional quadratic constraint on the system. The overall optimization problem is given by

$$\text{minimize} \quad J(F) = \mathbb{E}\left[\sum_{t \geq 0} x_t^\top Q_1 x_t + u_t^\top R_1 u_t\right],$$

$$\text{subject to} \quad D(F) = \mathbb{E}\left[\sum_{t \geq 0} x_t^\top Q_2 x_t + u_t^\top R_2 u_t\right] \leq D_0,$$

where $Q_1, Q_2, R_1$, and $R_2$ are positive definite matrices. Note that even thought the matrices are positive definite, both the objective function $J$ and the constraint $D$ are nonconvex with respect to

the parameter $F$. Furthermore, with the additional constraint, the optimal control sequence may no longer be linear in the state $x_t$. Nevertheless, in this work, we still consider linear control given by (13) and the goal is to find the best linear control for this constrained LQR problem. We assume that the choice of $A, B$ are such that the optimal cost is finite.

**Random initial state.** We first consider the setting where the initial state $x_0 \sim \mathcal{D}$ follows a random distribution $\mathcal{D}$, while both the state transition and the control sequence (13) are deterministic (i.e. $v_t = w_t = 0$). In this random initial state setting, [24] showed that without the constraint, the policy gradient method converges efficiently to the global optima in polynomial time. In the constrained case, we can explicitly write down the objective and constraint function, since the only randomness comes from the initial state. Therefore, we have the state dynamic $x_{t+1} = (A - BF)x_t$ and the objective function has the following expression ([24], Lemma 1)

$$J(F) = \mathbb{E}_{x_0 \sim \mathcal{D}} \left[ x_0^\top P_F x_0 \right], \tag{14}$$

where $P_F$ is the solution to the following equation

$$P_F = Q_1 + F^\top R_1 F + (A - BF)^\top P_F (A - BF). \tag{15}$$

The gradient is given by

$$\nabla_F J(F) = 2 \Big( \big( R_1 + B^\top P_F B \big) F - B^\top P_F A \Big) \cdot \left[ \mathbb{E}_{x_0 \sim \mathcal{D}} \sum_{t=0}^{\infty} x_t x_t^\top \right]. \tag{16}$$

Let $S_F = \sum_{t=0}^{\infty} x_t x_t^\top$ and observe that

$$S_F = x_0 x_0^\top + (A - BF) S_F (A - BF)^\top. \tag{17}$$

We start from some $F_0$ and apply our Algorithm 1 to solve the constrained LQR problem. In iteration $k$, with the current estimator denoted by $F_k$, we first obtain an estimator of $P_{F_k}$ by starting from $Q_1$ and iteratively applying the recursion $P_{F_k} \leftarrow Q_1 + F_k^\top R_1 F_k + (A - BF_k)^\top P_{F_k} (A - BF_k)$ until convergence. Next, we sample an $x_0^*$ from the distribution $\mathcal{D}$ and follow a similar recursion given by (17) to obtain an estimate of $S_{F_k}$. Plugging the sample $x_0^*$ and the estimates of $P_{F_k}$ and $S_{F_k}$ into (14) and (16), we obtain the sample reward value $J^*(F_k)$ and $\nabla_F J^*(F_k)$, respectively. With these two values, we follow (5) and (8) and obtain $\overline{J}^{(k)}(F)$. We apply the same procedure to the cost function $D(F)$ with $Q_1, R_1$ replaced by $Q_2, R_2$ to obtain $\overline{D}^{(k)}(F)$. Finally we solve the optimization problem (7) (or (9) if (7) is infeasible) and obtain $F_{k+1}$ by (10).

**Random state transition and control.** We then consider the setting where both $v_t$ and $w_t$ are independent standard Gaussian white noise. In this case, the state dynamic can be written as $x_{t+1} = (A - BF)x_t + \epsilon_t$ where $\epsilon_t \sim \mathcal{N}(0, I + BB^\top)$. Let $P_F$ be defined as in (15) and $S_F$ be the solution to the following Lyapunov equation

$$S_F = I + BB^\top + (A - BF) S_F (A - BF)^\top.$$

The objective function has the following expression ([68], Proposition 3.1)

$$J(F) = \mathbb{E}_{x \sim \mathcal{N}(0, S_F)} \left[ x^\top (Q_1 + F^\top R_1 F) x \right] + \text{tr}(R_1), \tag{18}$$

and the gradient is given by

$$\nabla_F J(F) = 2 \Big( \big( R_1 + B^\top P_F B \big) F - B^\top P_F A \Big) \cdot \mathbb{E}_{x \sim \mathcal{N}(0, S_F)} \left[ xx^\top \right]. \tag{19}$$

Although in this setting it is straightforward to calculate the expectation in a closed form, we keep the current expectation form to be in line with our algorithm. Moreover, when the error distribution is more complicated or unknown, we can no longer calculate the closed form expression and have to sample in each iteration. With the formulas given by (18) and (19), we again apply our Algorithm 1. We sample $x \sim \mathcal{N}(0, S_F)$ in each iteration and solve the optimization problem (7) or (9). The whole procedure is similar to the random initial state case described above.

**Other applications.** Our algorithm can also be applied to constrained parallel MDP and constrained multi-agent MDP problem. Due to the space limit, we relegate them to supplementary material.

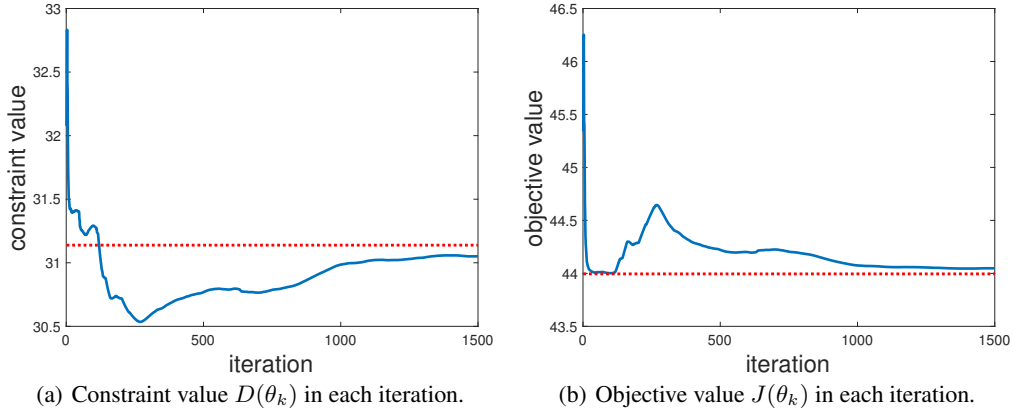

(a) Constraint value $D(\theta_k)$ in each iteration.  (b) Objective value $J(\theta_k)$ in each iteration.

Figure 1: An experiment on constrained LQR problem. The iterate starts from an infeasible point and then becomes feasible and eventually converges.

| | min value | # iterations | approx. min value | approx. # iterations |
|---|---|---|---|---|
| Our method | $30.689 \pm 0.114$ | $2001 \pm 1172$ | $30.694 \pm 0.114$ | $604.3 \pm 722.4$ |
| Lagrangian | $30.693 \pm 0.113$ | $7492 \pm 1780$ | $30.699 \pm 0.113$ | $5464 \pm 2116$ |

Table 1: Comparison of our method with Lagrangian method

## 6 Experiment

We verify the effectiveness of the proposed algorithm through experiments. We focus on the LQR setting with a random initial state as discussed in Section 5. In this experiment we set $x \in \mathbb{R}^{15}$ and $u \in \mathbb{R}^{8}$. The initial state distribution is uniform on the unit cube: $x_0 \sim \mathcal{D} = \text{Uniform}\big([-1,1]^{15}\big)$. Each element of $A$ and $B$ is sampled independently from the standard normal distribution and scaled such that the eigenvalues of $A$ are within the range $(-1,1)$. We initialize $F_0$ as an all-zero matrix, and the choice of the constraint function and the value $D_0$ are such that (1) the constrained problem is feasible; (2) the solution of the unconstrained problem does not satisfy the constraint, i.e., the problem is not trivial; (3) the initial value $F_0$ is not feasible. The learning rates are set as $\eta_k = \frac{2}{3}k^{-3/4}$ and $\rho_k = \frac{2}{3}k^{-2/3}$. The conservative choice of step size is to avoid the situation where an eigenvalue of $A - BF$ runs out of the range $(-1,1)$, and so the system is stable. [5]

Figure 1(a) and 1(b) show the constraint and objective value in each iteration, respectively. The red horizontal line in Figure 1(a) is for $D_0$, while the horizontal line in Figure 1(b) is for the unconstrained minimum objective value. We can see from Figure 1(a) that we start from an infeasible point, and the problem becomes feasible after about 100 iterations. The objective value is in general decreasing after becoming feasible, but never lower than the unconstrained minimum, as shown in Figure 1(b).

**Comparison with the Lagrangian method.**   We compare our proposed method with the usual Lagrangian method. For the Lagrangian method, we follow the algorithm proposed in [18] for safe reinforcement learning, which iteratively applies gradient descent on the parameter $F$ and gradient ascent on the Lagrangian multiplier $\lambda$ for the Lagrangian function until convergence.

Table 1 reports the comparison results with mean and standard deviation based on 50 replicates. In the second and third columns, we compare the minimum objective value and the number of iterations to achieve it. We also consider an approximate version, where we are satisfied with the result if the objective value exceeds less than 0.02% of the minimum value. The fourth and fifth columns show the comparison results for this approximate version. We can see that both methods achieve similar minimum objective values, but ours requires less number of policy updates, for both minimum and approximate minimum version.

## Footnotes

[5]The code is available at `https://github.com/ming93/Safe_reinforcement_learning`

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
