[Supplementary Material]

# A    Other applications

## A.1    Constrained Parallel Markov Decision Process

We consider the parallel MDP problem [34, 42, 17] where we have a single-agent MDP task and $N$ workers, where each worker acts as an individual agent and aims to solve the *same* MDP problem. In the parallel MDP setting, each agent is characterized by a tuple $(\mathcal{S}, \mathcal{A}, P, \gamma, r^i, d^i, \mu^i)$, where each agent has the same but individual state space, action space, transition probability distribution, and the discount factor. However, the reward function, cost function, and the distribution of the initial state $s_0 \in \mathcal{S}$ could be different for each agent, but satisfy $\mathbb{E}[r^i(s,a)] = r(s,a)$, $\mathbb{E}[d^i(s,a)] = d(s,a)$, and $\mathbb{E}[\mu^i(s,a)] = \mu(s,a)$. Each agent $i$ generates its own trajectory $\{s_0^i, a_0^i, s_1^i, a_1^i, \dots\}$ and collects its own reward/cost value $\{r_0^i, d_0^i, r_1^i, d_1^i, \dots\}$.

The hope is that by solving the single-agent problem using $N$ agents in parallel, the algorithm could be more stable and converge much faster [40]. Intuitively, each agent $i$ may have a different initial state and will explore different parts of the state space due to the randomness in the state transition distribution and the policy. It also helps to reduce the correlation between agents' behaviors. As a result, by running multiple agents in parallel, we are more likely to visit different parts of the environment and get the experience of the reward/cost function values more efficiently. This mimics the strategy used in tree-based supervised learning algorithms [14, 29, 30].

Following the settings in [17], we have $N$ agents (i.e., $N$ workers) and one central controller in the system. The global parameter is denoted by $\theta$, and we consider the constrained parallel MDP problem where the goal is to solve the following optimization problem:

$$\underset{\theta}{\text{minimize}} \quad J(\theta) = \sum_{i=1}^{N} \mathbb{E}_{\pi_\theta}\left[-\sum_{t \geq 0} \gamma^t \cdot r^i(s_t^i, a_t^i)\right],$$

$$\text{subject to} \quad D(\theta) = \mathbb{E}_{\pi_\theta}\left[\sum_{t \geq 0} \gamma^t \cdot d^i(s_t^i, a_t^i)\right] \leq D_0, \quad i \in \mathcal{N}.$$

During the estimation step, the controller broadcasts the current parameter $\theta_k$ to each agent and each agent samples its own trajectory and obtains estimators for function value/gradient of the reward/cost function. Next, each agent uploads its estimators to the central controller and the central controller takes the average over these estimators, and then follow our proposed algorithm to solve for the QCQP problem and update the parameter to $\theta_{k+1}$. This process continues until convergence.

## A.2    Constrained Multi-agent Markov Decision Process

A natural extension of the (single-agent) MDP is to consider a model with $N$ agents termed multi-agent Markov decision process (MMDP). Recently this kind of problem has been attracting more and more attention. See [15] for a comprehensive survey. Most of the work on multi-agent MDP problems consider the setting where the agents share the same global state space, but each with their own collection of actions and rewards [11, 63, 69]. In each stage of the system, each agent observes the global state and chooses its own action individually. As a result, each agent receives its reward and the state evolves according to the joint transition distribution. An MMDP problem can be fully collaborative where all the agents have the same goal, or fully competitive where the problem consists of two agents with an opposite goal, or the mix of the two.

Here we consider a slightly different setting where each agent has its own state space. The only connection between the agents is that the global reward is a function of the overall states and actions. Furthermore, each agent has its own constraint which depends on its own state and action only. This problem is known as Transition-Independent Multi-agent MDP and is considered in [50]. Specifically, each agent's task is characterized by a tuple $(\mathcal{S}^i, \mathcal{A}^i, P^i, \gamma, d^i, \mu^i)$ with each component defined as usual. Note that $P^i \colon \mathcal{S}^i \times \mathcal{A}^i \to \mathcal{P}(\mathcal{S}^i)$ and $d^i \colon \mathcal{S}^i \times \mathcal{A}^i \to \mathbb{R}$ are functions of each agent's state and action only and do not depend on other agents. Denote $\mathcal{S} = \Pi_{i \in \mathcal{N}} \mathcal{S}^i$ and $\mathcal{A} = \Pi_{i \in \mathcal{N}} \mathcal{A}^i$ as the joint state space and action space. The global reward function is given by $r \colon \mathcal{S} \times \mathcal{A} \to \mathbb{R}$ that depends on the joint state and action. Here we consider the fully collaborative setting where all the agents have the same goal. Under this setting, the policy set of each agent is parameterized as $\{\pi_{\theta^i}^i \colon \mathcal{S}^i \to \mathcal{P}(\mathcal{A}^i)\}$ and we denote $\theta = [\theta^1, \dots, \theta^N]$ as the overall parameters and $\pi_\theta$ as the overall policy. In the following, we use $\mathcal{N} = \{1, 2, \dots, N\}$ to denote the $N$ agents. Denote $a_t^i$ as the action

chosen by agent $i$ at stage $t$ and $a_t = \Pi_{i \in \mathcal{N}} \, a_t^i$ as the joint action chosen by all the agents. The goal of this constrained MMDP is to solve the following problem

$$\underset{\theta}{\text{minimize}} \quad J(\theta) = \mathbb{E}_{\pi_\theta} \left[ -\sum_{t \geq 0} \gamma^t \cdot r(s_t, a_t) \right],$$

$$\text{subject to} \ \ D^i(\theta^i) = \mathbb{E}_{\pi_{\theta^i}} \left[ \sum_{t \geq 0} \gamma^t \cdot d^i(s_t^i, a_t^i) \right] \leq D_0^i, \ \ i \in \mathcal{N}. \tag{20}$$

Inspired by the parallel implementation ([38], Section V), our algorithm applies naturally to constrained MMDP problem with some modifications. This modified procedure can also be viewed as a distributed version of the original algorithm. The overall problem (20) can be viewed as a large "single-agent" problem where the constraints are decomposable into $N$ parts. In this case, instead of solving a large QCQP problem in each iteration, each agent could solve its own QCQP problem in a distributed manner which is much more efficient. As before, we denote the sample negative reward and cost function as

$$J^*(\theta) = -\sum_{t \geq 0} \gamma^t \cdot r(s_t, a_t) \qquad \text{and} \qquad D^{i,*}(\theta^i) = \sum_{t \geq 0} \gamma^t \cdot d^i(s_t^i, a_t^i).$$

In each iteration with $\theta_k = [\theta_k^1, ..., \theta_k^N]$, we approximate $J(\theta)$ and $D(\theta)$ as

$$\widetilde{J}^i(\theta^i, \theta_k, \tau) = \frac{1}{N} J^*(\theta_k) + \langle \nabla_{\theta^i} J^*(\theta_k), \theta^i - \theta_k^i \rangle + \tau \|\theta^i - \theta_k^i\|_2^2,$$

$$\widetilde{D}^i(\theta^i, \theta_k, \tau) = D^{i,*}(\theta_k^i) + \langle \nabla_{\theta^i} D^{i,*}(\theta_k^i), \theta^i - \theta_k^i \rangle + \tau \|\theta^i - \theta_k^i\|_2^2.$$

Note that the constraint function is naturally decomposable into $N$ parts. We also "manually" split the objective function into $N$ parts, so that each agent could solve its own QCQP problem in a distributed manner. As before, we define

$$\overline{J}^{i,(k)}(\theta^i) = (1 - \rho_k) \cdot \overline{J}^{i,(k-1)}(\theta^i) + \rho_k \cdot \widetilde{J}^i(\theta^i, \theta_k, \tau),$$

$$\overline{D}^{i,(k)}(\theta^i) = (1 - \rho_k) \cdot \overline{D}^{i,(k-1)}(\theta^i) + \rho_k \cdot \widetilde{D}^i(\theta^i, \theta_k, \tau).$$

With this surrogate functions, each agent then solves its own convex relaxation problem

$$\overline{\theta}_k^i = \underset{\theta^i}{\text{argmin}} \ \overline{J}^{i,(k)}(\theta^i) \qquad \text{subject to} \qquad \overline{D}^{i,(k)}(\theta^i) \leq D_0^i, \tag{21}$$

or, alternatively, solves for the feasibility problem if (21) is infeasible

$$\overline{\theta}_k^i = \underset{\theta^i, \alpha^i}{\text{argmin}} \ \alpha^i \qquad \text{subject to} \qquad \overline{D}^{i,(k)}(\theta^i) \leq D_0^i + \alpha^i.$$

This step can be implemented in a distributed manner for each agent and is more efficient than solving the overall problem with the overall parameter $\theta$. Finally, the update rule for each agent $i$ is as usual

$$\theta_{t+1}^i = (1 - \eta_k) \cdot \theta_k^i + \eta_k \cdot \overline{\theta}_k^i.$$

This process continues until convergence.

## B  Proof of Theorem 4

According to the choice of the surrogate function (5) and Assumption 2, it is straightforward to verify that the function $\overline{J}^{(k)}(\theta)$ defined in (8) is uniformly strongly convex in $\theta$ for each iteration $t$. Moreover, both $\overline{J}^{(k)}(\theta)$ and $\nabla_\theta \overline{J}^{(k)}(\theta)$ are Lipschitz continuous functions.

From Lemma 1 in [49] we have

$$\lim_{t \to \infty} \left| \overline{J}^{(k)}(\theta) - \mathbb{E}\big[\widetilde{J}(\theta, \theta_k, \tau)\big] \right| = 0.$$

Since the function $\mathbb{E}\big[\widetilde{J}(\theta, \theta_k, \tau)\big]$ is Lipschitz continuous in $\theta_k$, we obtain that

$$\left| \overline{J}^{(k_1)}(\theta) - \overline{J}^{(k_2)}(\theta) \right| \leq L_0 \cdot \|\theta_{k_1} - \theta_{k_2}\| + \epsilon,$$

for some constant $L_0$ and the error term $\epsilon$ that goes to 0 as $k_1, k_2$ go to infinity. This shows that the function sequence $\overline{J}^{(k_j)}(\theta)$ is equicontinuous. Since $\Theta$ is compact and the discounted cumulative reward function is bounded by $r_{\max}/(1-\gamma)$, we can apply Arzela-Ascoli theorem [22, 32] to prove existence of $\widehat{J}(\theta)$ that converges uniformly. Moreover, since we apply the same operations on the constraint function $D(\theta)$ as to the reward function $J(\theta)$ in Algorithm 1, the above properties also hold for $D(\theta)$.

The rest of the proof follows in a similar way as the proof of Theorem 1 in [38]. Under Assumptions 1 - 3, the technical conditions in [38] are satisfied by the choice of the surrogate functions (5) and (6). According to Lemma 2 in [38], with probability one we have

$$\limsup_{k \to \infty} D(\theta_k) \leq D_0.$$

This shows that, although in some of the iterations the convex relaxation problem (7) is infeasible, and we have to solve the alternative problem (9), the iterates $\{\theta_k\}$ converge to the feasible region of the original problem (4) with probability one. Furthermore, with probability one, the convergent point $\widetilde{\theta}$ is the optimal solution to the following problem

$$\underset{\theta \in \Theta}{\text{minimize}} \quad \widehat{J}(\theta) \qquad \text{subject to} \qquad \widehat{D}(\theta) \leq D_0. \tag{22}$$

The KKT conditions for (22) together with the Slater condition show that the KKT conditions of the original problem (4) are also satisfied at $\widetilde{\theta}$. This shows that $\widetilde{\theta}$ is a stationary point of the original problem almost surely.