[Reviews · NeurIPS 2019]

Reviewer 1



post-rebuttal update: I'm increasing my score from 5 -> 6. My main concerns were: - novelty WRT [34]: I didn't update on this; it doesn't seem very novel, but that's not a deal-breaker. - experiments: The new results are more convincing (but still could be stronger). Quality 4 -> 5 Overall 5 -> 6 Overall, this seems like a nice paper, but I found it hard to evaluate given my background. I also with the authors had given some intuition for the theoretical properties of their method. My main concerns are over the originality (it seems very similar to [34]), and the weakness of the experiments. Originality: 5/10 This paper seems mostly to be about transferring the more general result of [34] to the specific setting of constrained MDPs. So I wish the authors gave more attention to [34], specifically: - reviewing the contribution of [34] in more detail - clarifying the novelty of this work (Is it in the specific design choices? The actor-critic algorithm? The set-up in lines 532-542 (Appendix)? Is it just in noticing the suitability of [34]'s results for CMDPs?) Without this help from the authors, it's difficult for me assess the originality and significance of their work. At the moment, it looks to me like a pretty straightforward application of [34]s results to CMDPs. Quality: 4/10 Overall the paper seems technically sound and well-done. But the experiments seem like an after-thought and/or purely illustrative, since: 1) they aren't included in the main paper and 2) they don't include multiple trials for significance. I also didn't find the content in section 5 particularly valuable; I think experiments would have been better. I'm also not familiar with the setting (LQR), or previous work using RL for LQR (with or without constraints), and the authors' summary didn't give me enough context to interpret the strength or meaning of their results. The abstract and introduction also over-claim, stating that they "apply" the algorithm to multi-agent RL; but there are no experiments for that setting, only description of how it could be applied. Clarity: 7/10 The paper is mostly quite clearly written, with a few typos. The following should be clarified: - the novelty compared with [34] - the significance of the experimental results - the main baseline being methods with Lagrange-multipliers - the motivations from safety: i.e. this doesn't tackle safe exploration, right? It's for safe execution? How exactly would this kind of method be used to make a safe real-world system? What problems does(n't) it solve? For instance, is it possible to verify that constraints will be satisfied? In practice, do CMDP approaches out-perform incorporating constraints as large negative rewards (when)? Also, *some* intuition for the proof should be given. This should include both the intuition underlying the proof strategy from [34], and an explanation highlighting the differences and the explaining what the issues are the require modifications to their proof. Significance: 7/10 The introduction presents a strong case that (in some theoretical aspects) this work represents a significant step up from the standard approach to CMDPs, based on Lagrangian multipliers. A table summarizing different approaches to CMDPs and highlighting the advantages of this work over previous approaches might be a nice addition. However, without stronger experiments, I don't think the paper presents a strong case that this method will be superior to existing methods in practice. Some detailed suggestions: - Line 85: "minimize (1)" --> "maximize (1)" - Line 109-110 and line 83 are redundant - Algorithm 2: beta_w/v are not defined; initial values of delta^J/D are not defined - line 201: "and their are" --> "and their derivatives are" (I think?) - line 261 (right hand side): PF_k --> P_{F_k} - I suggest renaming Q1, R1, Q2, R2 as Q_J, R_J, Q_D, R_D Some Questions: - Appendix: what's the difference between figure 1a/b and figure 2a/b? - Line 152-153: why can a solution be written in closed form? (explain or reference). - Line 186: how/why does actor-critic "improve the performance further"? And is the improvement in terms of performance, rate of convergence, both, neither? And how is the claim (of improved performance) supported? - is finding a stationary point good enough? why can't we do better? (e.g. local minima?) - What is the state of RL research on LQR? How does it compare with other approaches? Is studying RL algorithms in LQR interesting practically, scientifically, or both? Some high level questions: - what are the key contributions of this work compared to [34]? - what are the key challenges for the proof? - what is the point of section 5? [34] An Liu, Vincent Lau, and Borna Kananian. Stochastic successive convex approximation for non-convex constrained stochastic optimization. arXiv preprint arXiv:1801.08266, 2018

Reviewer 2



-- The rebuttal answered all my questions. -- The problem that this paper addresses is to minimize or maximize an objective by optimizing a policy that is parameterized by a nonlinear function approximation subject to a set of constraints. The general parametrization is nonconvex, so the objective and the constraints are given by nonconvex functions, and additionally they are also stochastic. The authors propose constructing a convex approximation to the nonconvex functions by using a first-order approximation and an additional strong convex term to ensure stability and the convergence of the method. The authors leverage previous results from [34] to prove that their method converges to a stationary point as the number of iterations approaches to infinity. A constrained linear-quadratic regulator problem is used to show the applicability of their method. Detailed comments: Introduction: The authors give a good overview of the state-of-the-art in policy optimization with respect to constraints. Particularly, they compare their approach to Lagrangian methods, which relax the constraint by using a linear penalty function, and then solve a reinforcement learning problem with a given Lagranian multiplier. They also mention a cross-entropy-based optimization method that achieves asymptotic convergence. Theoretical section: In line 85, I think the authors mean "maximize (1)" instead of "minimize (1)", given the optimization problem in Equation (1). In line 94, the authors mention further reducing the variance without giving any prior explanation, and it was confusing to me. In line 119, the authors replace the objective and the constraint functions by surrogate functions with "nice" properties, and do not explain what is a nice property. The authors state some properties such as strong convexity, and they will use a strong convex function, but do not motivate the properties of the surrogate functions that they have used. In line 137, it could be useful to define the gradient of the objective and the constraint function, or give a reference to the policy gradient theorem, the gradient of the functions are undefined in the paper at the moment. In (5) and (6), the authors just state that \tau is positive, but they do mention the value of \tau anywhere in the paper, including in the main theorem or in the supplementary materials. I am not sure if the expression for the J_truncate after line 173 is correct. For instance, suppose that \gamma is very close to 1, this means that the expected value of the trajectory would be very high, but the truncated value according to the expression would be very small. For the main theoretical result, I think the Assumption 3 may be too strong, it basically states that any stationary point of the optimization problem is feasible, and makes sure that the procedure can improve on the infeasibility. However, I believe that, the assumption is too strong, and it basically guarantees that the problem is always solvable with any kind of method. Evaluation: The authors compare their approach with a Lagrangian method on a constrained linear-quadratic regulator problem, and show that their method achieves feasibility and converges to a stationary point faster than a Lagrangian method. I think the authors should also compare their approach to the cross-entropy method. The authors state that their approaches work for general constrained Markov decision processes with multiple agents but they give no numerical evaluation for these problems. No code is provided with the supplementary materials.

Reviewer 3



Overall, the paper is solving an important problem, and provides compelling convergence guarantees. However, my main question is exactly why the constrained RL setting is more challenging (from a theoretical perspective) compared to constrained optimization. In particular, the authors’ approach seems to be based heavily on [34], but they do not clearly distinguish why [34] cannot be directly applied to solving a CMDP problem. I would be much more convinced about the novelty of this paper if such a discussion is provided. Furthermore, the proposed algorithm does not appear to guarantee safety during the training phase. Assuming I am understanding correctly, the authors should make it clear that their goal is to identify a safe policy rather than ensure safety during learning.

[Author Response · NeurIPS 2019]

We thank the reviewers for their comments. We will fix the typos in the final version.

**Comparison with [34]:** In general solving the safe RL problem is a harder problem. In order to apply to Reinforce-
ment learning, we need to verify that the Lipschitz condition is satisfied, and also the policy gradient has to be estimated
(instead of directly evaluated as in a standard optimization problem). The usage of Actor-Critic algorithm reduces the
variance of the sampling a lot (see next paragraph), which is unique to Reinforcement learning.

**Actor-Critic improves performance and reduces variance:** In general, an actor-critic method helps reduce variance
by using a value function instead of Monte-Carlo sampling. Specifically, in Algorithm 1 we need to obtain a sample
trajectory and calculate $J^*(\theta)$ and $\nabla_\theta J^*(\theta)$ by Monte-Carlo sampling. This step has a high variance since we need
to sample a potentially long trajectory and sum up a lot of random rewards. In contrast, in Algorithm 2, this step is
replaced by a value function $V_w^J(s)$, which reduces the variance. We will clarify this in the final version.

**About safe exploration:** Our algorithm does not guarantee safe exploration during the training phase. Ensuring
safety during learning is a more challenging problem. Sometimes even finding a feasible point is not straightforward,
otherwise Assumption 3 is not necessary. We will clarify this in final version, and leave safety training to future work.

**Reviewer 1:** Experiments: we run additional multiple trials to compare our method with Lagrangian method. The
following table reports the averaged results with mean and standard deviation. In columns 2 and 3, we compare the
minimum value and number of iterations to achieve it. Since for both the methods, the constraint values oscillate above
and below $D_0$ (as shown in Figure 2cd), we also consider an approximate version, where we are satisfied with the result
if the objective value exceeds less than 0.2% of the minimum value. Columns 4, 5 of the table report the averaged
results for this approximate version. We can see that both methods achieve similar minimum values, but ours requires
less number of policy updates, for both minimum and approximate minimum version. We will include this experiment
result, and move all the experiment parts to the main text in the final version, since we have an additional page.

|  | minimum value | # iterations | approximate minimum value | approximate # iterations |
|---|---|---|---|---|
| Our method | $30.689 \pm 0.114$ | $2001 \pm 1172$ | $30.694 \pm 0.114$ | $604.3 \pm 722.4$ |
| Lagrangian | $30.693 \pm 0.113$ | $7492 \pm 1780$ | $30.699 \pm 0.113$ | $5464 \pm 2116$ |

Other comments: **(i).** Line 201: yes, it should be "and their derivatives are". **(ii).** Figure 1a/b and Figure 2a/b are two
realizations under the same setting. It is safe to ignore Figure 1a/b. **(iii).** Note that we are dealing with nonconvex
optimization with a nonconvex constraint. When adding random noise to the iterates, we could escape saddle points
and achieve a second-order stationary point. **(iv).** Line 152 closed form: Solving (9) is effectively finding a minimum
value of $\overline{D}^{(k)}(\theta)$, and $\alpha = \min_\theta \overline{D}^{(k)}(\theta) - D_0$. Looking at (6) and (8), $\overline{D}^{(k)}(\theta)$ is quadratic and decomposable to each
component of $\theta$. Therefore it is minimizing several quadratic functions, and we have closed form solution. **(v).** Section
5 provides an example of concrete application of our approach. The settings are used in experiments. **(vi).** To verify
the constraint is satisfied, we could use Monte-Carlo sampling which could be computationally heavy. **(vii).** Even for
simple problems, modeling the constraints as negative rewards will lead to very conservative or risky behavior. See
[Undurti, Aditya. Planning under uncertainty and constraints for teams of autonomous agents. 2011]. **(viii).** Intuition
for the proof: we first show that the surrogate functions converge (Line 218), and then show that the iterates converge to
feasible region (Line 545). Using Slater's condition completes the proof.

As defined in Section 5, LQR is a control problem, and hence naturally fits into RL. Standard LQR has closed form
solution. One research direction is on algorithm part, to modify the problem so that it no longer has a closed form
solution. Then RL algorithm is useful to solve this modified problem, for example in this paper with safety constraint.
Another is to understand the convergence of RL algorithms on LQR, see [22] for policy gradient, and [7] for actor-critic.

**Reviewer 2: (i).** L119: we will provide more intuitions and references on the surrogate functions. **(ii).** L137: the
definition and reference of policy gradient algorithm is given in L92-93 in background section. We will provide a
reference to equation (3) here in the final version. **(iii).** In (5) and (6), $\tau$ can be any positive constant and it does not
affect the theoretical convergence. **(iv).** Expression for the $J_{truncate}$: if $\gamma$ is close to 1, the random variable $T$ would be
large. Therefore the summation of the reward $\sum_{t=0}^T r(s_t, a_t)$ would be large. The term $(1 - \gamma)$ is like a normalizer.
The intuition is that, if $\gamma$ is close to 1, then future rewards are important so we need a large $T$. This is captured by the
definition of $T$ in L172. **(v).** Experiments: we will release codes on github and evaluate on other general set of problem
in the final version. The paper on cross entropy method [59] does not have codes so we only compare with the standard
(and easy to implemented) Lagrangian method.

Assumption 3 is indeed a relatively strong assumption. As discussed in line 222, it is not necessary if we initialize
with a feasible point. Moreover, if (in practice) we reach a feasible point in the iterates, then we could view it as an
initializer as again Assumption 3 is not necessary. If we could not find a feasible point, then the iterates may converge
to an infeasible stationary point of (12). As far as we know, without Assumption 3 we can not rule out this case. In
practice, we could initialize with multiple start points, and the convergence is then guaranteed as long as we reach a
single feasible point for one of these iterates. For our experiments on LQR, for every single replicate, we could reach a
feasible point, and therefore Assumption 3 is not necessary.

[Meta-Review · NeurIPS 2019]

The reviewers found that the problem addressed in this paper is interesting. While they had some concerns regarding the overlap with prior work, these concerns were mostly addressed in the rebuttal and some reviewers therefore raised their score.